# Transcriptome Profile Analysis of Triple-Negative Breast Cancer Cells in Response to a Novel Cytostatic Tetrahydroisoquinoline Compared to Paclitaxel

**DOI:** 10.3390/ijms22147694

**Published:** 2021-07-19

**Authors:** Madhavi Gangapuram, Elizabeth A. Mazzio, Kinfe K. Redda, Karam F. A. Soliman

**Affiliations:** Pharmaceutical Sciences Division, College of Pharmacy & Pharmaceutical Sciences, Institute of Public Health, Florida A&M University, Tallahassee, FL 32307, USA; madhavi.gangapuram@famu.edu (M.G.); elizabeth.mazzio@famu.edu (E.A.M.)

**Keywords:** drug discovery, cytostatic, cancer

## Abstract

The absence of chemotherapeutic target hormone receptors in breast cancer is descriptive of the commonly known triple-negative breast cancer (TNBC) subtype. TNBC remains one of the most aggressive invasive breast cancers, with the highest mortality rates in African American women. Therefore, new drug therapies are continually being explored. Microtubule-targeting agents such as paclitaxel (Taxol) interfere with microtubules dynamics, induce mitotic arrest, and remain a first-in-class adjunct drug to treat TNBC. Recently, we synthesized a series of small molecules of substituted tetrahydroisoquinolines (THIQs). The lead compound of this series, with the most potent cytostatic effect, was identified as 4-Ethyl-*N*-(7-hydroxy-3,4-dihydroisoquinolin-2(1*H*)-yl) benzamide (GM-4-53). In our previous work, GM-4-53 was similar to paclitaxel in its capacity to completely abrogate cell cycle in MDA-MB-231 TNBC cells, with the former not impairing tubulin depolymerization. Given that GM-4-53 is a cytostatic agent, and little is known about its mechanism of action, here, we elucidate differences and similarities to paclitaxel by evaluating whole-transcriptome microarray data in MDA-MB-231 cells. The data obtained show that both drugs were cytostatic at non-toxic concentrations and caused deformed morphological cytoskeletal enlargement in 2D cultures. In 3D cultures, the data show greater core penetration, observed by GM-4-53, than paclitaxel. In concentrations where the drugs entirely blocked the cell cycle, the transcriptome profile of the 48,226 genes analyzed (selection criteria: (*p*-value, FDR *p*-value < 0.05, fold change −2< and >2)), paclitaxel evoked 153 differentially expressed genes (DEGs), GM-4-53 evoked 243 DEGs, and, of these changes, 52/153 paclitaxel DEGs were also observed by GM-4-53, constituting a 34% overlap. The 52 DEGS analysis by String database indicates that these changes involve transcripts that influence microtubule spindle formation, chromosome segregation, mitosis/cell cycle, and transforming growth factor-β (TGF-β) signaling. Of interest, both drugs effectively downregulated “inhibitor of DNA binding, dominant negative helix-loop-helix” (ID) transcripts; ID1, ID3 and ID4, and amphiregulin (AREG) and epiregulin (EREG) transcripts, which play a formidable role in cell division. Given the efficient solubility of GM-4-53, its low molecular weight (MW; 296), and capacity to penetrate a small solid tumor mass and effectively block the cell cycle, this drug may have future therapeutic value in treating TNBC or other cancers. Future studies will be required to evaluate this drug in preclinical models.

## 1. Introduction

Breast cancer continues to be a major public health concern, particularly the subclass referred to as “triple-negative breast cancer” (TNBC). TNBC is characterized by an absence of endocrine chemotherapy receptor targets, fewer treatment options, and greater occurrence in premenopausal and African American women (AAW) [1]. In AAW, TNBC is typically associated with advanced-stage diagnosis, shorter disease-free survival, a proclivity toward distant bone metastasis, and higher mortality than non-TNBC breast cancers [2,3]. Standard treatments for TNBC include the combined use of cytotoxic and cytostatic agents; taxanes (docetaxel and paclitaxel), anthracyclines (doxorubicin and epirubicin), cyclophosphamide, fluorouracil, capecitabine, or platinum-based drugs [4,5]. Taxanes mediate cytostatic effects by holding the cytoskeletal architecture static. The cytoskeleton is made up of microtubules (alpha- and beta-heterodimers), where paclitaxel binds to tubulin and disrupts the dynamics of polymerization and depolymerization, which otherwise required for microtubule treadmilling and mitosis. Most of the research on taxanes explores how they mediate effects on tubulin dynamics rather than how they alter the cancer transcriptome, except for eventual drug resistance. However, cancer itself is a disease mired in pathological genetic alteration, which results in rampant cell proliferation.

Breast cancer is pathologically demarcated by a plethora of abnormal gene profiles and transcripts that drive tumor initiation (MYC, Erb-B2 Receptor tyrosine kinase 2 (ERBB2) [6,7], tumor progression (FOS, JUNB) [8], proliferation and metastasis (epidermal growth factor receptor (EGRF)) [9,10]. These changes are worsened by the concurrent loss of tumor suppressors (transcription factor p53, E-cadherin) [6,11] and global defects in DNA repair mechanisms (e.g., phosphatidylinositol 3,4,5-trisphosphate 3-phosphatase and dual-specificity protein phosphatase (PTEN), retinoblastoma protein 1 (RB1) and breast-cancer-associated (BRCA) 1,2 genes) [12,13,14].

The TNBC sub-class contains a confounding underlying pathological gene profile with a greater propensity for epithelial-to-mesenchymal transition (EMT), rapid cell proliferation (cyclin-dependent kinase inhibitor 1A (P21), PI3K/Akt, Wnt/beta-catenin signaling), and stem cell differentiation (S-100, p63, or vimentin), with most, if not all, abnormalities associated with underlying changes in epigenetic architecture [15,16,17,18,19,20]. Even worse, once TNBC malignancy is established, tertiary pathological gene profiles can arise from the treatments themselves, leading to lethal chemo- and radiation-resistant cancer. These include changes in gene expression profiles for chemokine and leukocyte recruitment peptides (PTGS2, IL-6, CCL2, CXCL8, and CXCL12) [21,22] those that drive angiogenesis (ANGPT1, VEGFA), cell-cycle progression proteins (EGR1, MYC, FOS, CDKN1A, CA2, ANKRD46) and those involved with multidrug resistance [15,21,23,24,25]. Chemoresistant tumors are circumscribed by advanced tumor-promoting pathways involving JAK/STAT3, [26] HSPC154, PI3K-Akt, [27,28] CYP1A1, TNF [29], TAZ-TEAD-Cyr61/CTGF [30], folate receptor 1 signaling (FOLR1) [31] and overexpression of paclitaxel resistance-associated genes (TRAGs), with many, if not all, being associated with underlying epigenetic-controlling elements [21,32]. The latter involves individual silenced hyper-methylated and hypo-methylated promoter regions of influential genes [33], and a fluctuation in miRNAs (e.g., miR-634), and/or circular RNAs (circRNAs) [34,35]. Even though mainline drugs used to treat TNBC, such as paclitaxel, in combination with anthracyclines, are largely effective, there are still limitations including side effects, chemoresistance, solubility, and delivery, with efforts being made to improve formulations or delivery systems [36].

Due to the great need for novel drug development in the area of TNBC, we synthesized a novel cytostatic compound; GM-4-53 (Figure 1), with similarities to paclitaxel in terms of therapeutic effect, without the well-known tubulin polymerization and thwarted retraction and treadmilling actions of the latter, which are the primary mechanism behind abrogated mitosis [37,38,39,40]. In this work, we further examine how paclitaxel and GM-4-53 affect the transcriptome of TNBC, to elucidate a plausible mechanism of action underpinning cytostatic effects [41].

## 2. Results

The anti-proliferative inhibitory growth (IG_50s_) was calculated by regression analysis from data acquired in a prolonged 6-day study, GM-4-53 (Figure 2A) and paclitaxel (Figure 2B). To establish that cell toxicity was not an interfering variable for both drugs, we conducted a 36-h toxicity assay over identical dose concentrations ranges at a higher plating density 0.5 × 10^5^ cells/well (data not shown), where no cytotoxic effects were found. Optimal concentrations for whole-transcriptomic (WT) microarray analysis were selected as 1 µg/mL for paclitaxel, and 5 µg/mL for GM-4-53, to establish a complete cell cycle blockade without cytotoxic effects, and a 36-h time point was chosen for endpoint analysis to elucidate changes at the gene transcript levels. The altered cytoskeletal changes evident at the 6-day endpoint are reflected in Figure 3, stained with phalloidin (actin) and propidium iodide (nuclear counterstain) in fixed permeabilized cells. The 10× images reflect the proliferation rate vs. controls, corresponding to the data in Figure 2A,B, where the 25× images show a zoom in on the gross abnormalities in cytoskeletal architecture evoked by both cytostatic agents. Furthermore, cell proliferation using comparative human female cancer cell lines was carried out by the Southern Research Institute, where the IG_50_s are presented in Table 1, showing efficacy in the mid-to-high *n*M range by GM-4-53 in diverse females cancer cell lines.

A summary view of significant DEG counts according to fold change, and significance (volcano plot) elicited by both drugs are presented in Figure 4. Specifics of these gene changes are presented in Figure 5, and the full data files can be found in Appendix A file, which lists gene symbol, gene description, average Log2 (signals) fold change (FC), *p*-Value, and false discovery rate (FDR). Appendix A contains information on each individual drug vs. control. Figure 5 summary shows that GM-4-53 evoke 75 upregulated DEGs vs. controls, 168 down-regulated DEGs vs. controls; paclitaxel evoked 108 upregulated DEGS vs. controls and 45 down-regulated DEGs vs. controls, with 52 of these genes being shared by both drugs. Of the 45 genes down-regulated by paclitaxel, 29 of these were also observed by GM-4-53, which constitutes a 64% overlap.

Using STRING/Protein-Protein Functional Enrichment Analysis, we further examine the 52 DEG overlap evoked by both paclitaxel and GM-4-53. (Figure 6) The data show both drug target genes having a primary molecular classification impact on spindle microtubules, chromosomal segregation, spindle microtubules to the kinetochore, TGF-beta signaling, and the MAPK signaling pathway. The 52-gene overlap, using a plotted fold overlay (Figure 7), shows the magnitude (FC) and direction (up/down) of these particular DEGs to be closely aligned; some of these are listed in Table 2.

The comparable efficacy of the two drugs in 3D tumor models was also established (Figure 8). Although a dose-response was carried out, a consistent morphological pattern of grossly distorted cytostatic cells was observed throughout all effective doses tested, as shown in Figure 2. The data in Figure 8 show a very distinct and unusual pattern between the two drugs, where paclitaxel appears to halt the growth of the spheroid and central core, whereas GM-4-53 halts the growth and appears to penetrate the tumor core, leading to mass dispersion of viable cytostatic cells. The ramifications of these findings need further investigation.

## 3. Discussion

The data in this work provide basic information on the transcriptome profile in response to the two cytostatic drugs (paclitaxel and GM-4-53) in a TNBC model. There is no doubt that GM-4-53 has many anticancer effects in common with paclitaxel, as both are cytostatic drugs, which fully block the cell cycle at non-cytotoxic concentrations, and evoke massive changes in the cytoskeletal architecture, disrupting microtubule dynamics and inducing cell-cycle arrest. Microtubule dynamics disrupters are mainline cancer drugs, as they can block the normal equilibrium, which allows cell structures to expand, retract, change shape, divide, and move. Cytostatic compounds such as paclitaxel are a front-line treatment in many cancers because they impair the dynamics of cytoskeletal fluidity in cancer cells, paralyzing microtubules (stabilized) by either blocking tubulin depolymerization or polymerization (destabilized), meaning that cell division is no longer possible [42,43]. This can occur via two basic classes of drugs (1) (e.g., taxanes, epothilones, cyclosstreptin, steroids, lactones, and natural compounds), referred to as microtubules stabilizing agents (MSA)s, which block tubulin depolymerization, or (2) drugs that inhibit polymerization of tubulin (destabilizing agents) (MDAs) such as colchicine, vinblastine and combretastatin-A4 [44]. MDAs and MSAs block cell-cycle progression holding the filaments static, thereby preventing chromosome attachment/segregation and spindle formation required for mitotic cell division [45,46]. The limitations to using these drugs to treat cancer, as is the case for paclitaxel, include inherent side effects, chemoresistance, and poor aqueous solubility, with limited access to the blood-brain barrier [47,48].

Drug therapeutic failures are ultimately responsible for mortality rates. As a result, there is a continuous demand for safe and efficient alternatives to TNBC cytostatic drugs, which may work through a different mechanism, enabling improved antitumor response and patient outcome. After our lab revealed GM-4-53 to be a potent cytostatic agent, the first obvious target evaluated was its effect on the tubulin polymerization/depolymerization processes; where the data failed to show any effect and showed only mild effects on the process of tubulin nucleation, suggesting there must be additional mechanisms in play [37].

Interestingly, while the effects of paclitaxel are not believed to involve a mechanism of action occurring at the gene level, the findings in this work were not expected, showing that many elements of the cell cycle are altered at the transcriptome level. It was also surprising to find significant overlap in the transcriptome in response to paclitaxel and GM-4-53, particularly in downregulated transcripts require for cell cycle and mitosis. Secondly, the data show DEG shifts that are unique to paclitaxel, which is reportedly observed to be associated with taxol chemoresistance, such as upregulated transcription of kinesin superfamily members, spindle assembly processes (e.g., MAD2, KIF11 (also known as kinesin-5 and Eg5), centrosomal proteins, centromere proteins, cell-division-cycle-associated genes, cyclins, centrioles, and aurora A, some of which also amplify chromosomal and spindle processes in paclitaxel refractory cancers [49,50,51,52] (see Appendix A). Information from whole-transcriptomic analysis on the effects of paclitaxel in this study may also provide information on changes that could align with drug resistance. While paclitaxel has been rigorously studied for decades, there is no known information on GM-4-53 or how it mediates its effect on cell division. For this reason, we searched for DEG overlaps in common to both drugs. Both drugs caused a pronounced downregulation of “inhibitor of DNA Binding/Inhibitor of differentiation” (ID) transcripts for ID1, ID3, and ID4, with GM-4-53 having more significant effects than paclitaxel. IDs are implicated in the control of cell division and mitosis not only during embryonic development but also in numerous cancers, including TNBC, being linked to larger tumor size, advanced histological grade, metastasis, vascular invasion, stem cell phenotype, lymph node invasion, and poor clinical outcomes [53,54]. The ID genes may control cell division by the indirect regulation of processes involving CDKN1A (p21) and CDKN1B (p27) [55]. In TNBC and other cancers, the ID oncogenic transcripts foster diverse cancer-related events, including EMT, signaling (e.g., EGFR/TGF-beta [56,57,58,59,60,61], K-Ras, WNT, STAT3, PI3K/Akt, OCT-4/ID1/NF-kappaB), where ID genes are a target of many anticancer drugs, including vinblastine [62,63]. Drugs such as GM-4-53 that can downregulate IDs are believed to offer therapeutic roles in the attenuation of TNBC progression and other cancers [64,65,66], with a capacity to offset chemoresistance associated with various drugs [67,68,69,70,71]. Regarding the impact of downregulated ID transcripts by both paclitaxel and GM-4-53, these are known to play a direct role in impeding cell division. Several studies seem to suggest that ID1 exerts control over cell cycle and self-renewal capacity of TNBC in vitro and in vivo [57], with its absence (silencing) leading to G0/G1 cell cycle arrest [56].

The data in this work also show that both paclitaxel and GM-4-53 co-downregulate both amphiregulin (AREG) and epiregulin (EREG) mRNAs, which are required for breast luminal development (by EGF binding/activating EGFR (ERBB)), and, when overexpressed, are linked to aggressive breast cancers of diverse type (ER + erb2, HER2 and TNBC) [72,73,74,75,76]. AREG/EREGs are also involved with enhanced tumor cell proliferation but equally capable of propelling the fibrotic processes required for the development of cancer-associated fibroblasts (CAFS), which establish a conducive tumor microenvironment (TME) for rapid tumor proliferation [77,78,79,80]. In brief, AREG/EREG ectodomain ligands are initially shed by the tumor necrosis factor-alpha-converting enzyme (TACE), with subsequent integration into exosomes. They are fully competent as ligands, which then bind/activate tumor EGFR (ErbB1-4) receptors to trigger oncogenic signaling: (phosphatidylinositol 3-kinase/Akt, Ras/Raf/MEK/ERK1/2, and phospholipase C), which also leads to sustained release of cytokines that cause leukocyte infiltration to the tumor microenvironment (TME) [81,82]. Additionally, GM-4-53 reduced the expression of Intercellular adhesion molecule 1 (ICAM-1), which is a driver of cell migration via the docking and trafficking of leukocytes toward cancer and stromal cells [83,84]. In addition, its intra-cellular product (pro-AREG) enters the nucleus, where it activates potent oncogenes [85,86,87,88]. The well-known tumor-promoting role of EGFT ligand/receptors is the premise behind existing chemotherapies [89], such as (cetuximab pertuzumab, trastuzumab), which bind ligands or interacts with the receptor (gefitinib, erlotinib, and lapatinib) [90]. Similar to EREG, drugs that interfere with AREG, particularly in chemoresistant breast cancers, effectively acquiesce tumor growth, tumor-associated macrophage (TAM) infiltration [91] and block the activation of diverse EGF receptors (ErbB1, 3 and 4+ ErbB2 HER2/Neu) [92,93,94,95].

While these are just a few promising aspects of GM-4-53, future work will be required to elucidate both the therapeutic and limiting factors of this drug. The data from this work clearly define the following: this drug is a cytostatic, non-cytotoxic agent with a similar effect on TNBC to the mainline drug, paclitaxel. The lack of cytotoxicity by paclitaxel has been observed in ex vivo explants of human breast tumors, which penetrate a tumor and disrupt mitosis without directly invoking cytotoxicity [96]. There is a clear distinction between cytotoxic drugs vs. cytostatic drugs, demarcated by the extremely large difference in drug dose concentrations between lower-dose effective blocks on cell proliferation (IG_50_) vs. cell viability ratios (IC_50_), where dead cells do not divide. While the data in this work also show that both drugs affect established 3D spheroid tumors, GM-453 appears to have significant effects on the clustering of cells or penetration of the tumor spheroid, which could theoretically either prevent tumor formation and/or alter the metastatic proclivity of circulating tumor cells [97].

## 4. Conclusions

In this work, we show that GM-4-53 displays several areas of potential targeted chemotherapy efficacy. Given the efficient solubility of GM-4-53, its low molecular weight (MW; 296), ability to penetrate a small solid tumor mass and effectively block the cell cycle, this drug may have future therapeutic value in treating TNBC or other cancers. Further research will be needed, including in vitro and in vivo preclinical work to establish limitations, safety guidelines, pharmacokinetics, and potential applications of this drug.

## 5. Materials and Methods

Hanks balanced salt solution (HBSS), (4-(2-hydroxyethyl)-1-piperazine-ethanesulfonic acid) (HEPES), ethanol, 96-well plates, flasks, paclitaxel, general reagents, and supplies were purchased from Sigma-Aldrich Co. (St. Louis, MO, USA) and VWR International (Radnor, PA, USA).

### 5.1. Chemistry

The synthesis and characterization of tetrahydroisoquinoline GM-4-53 have been described previously [37]. O-mesitylene sulfonyl hydroxylamine (MSH) was used to prepare the *N*-amino salt as an aminating agent, as previously reported [36]. MSH (22.74 mmol) was added, in dry methylene chloride (10 mL), an ice-cool solution of 7-hydroxyisoquinoline (20.67 mmol) in anhydrous methylene chloride and anhydrous methanol (1:1) (60 mL), over 5 min with stirring. The reaction was stirred at 0 °C for 6 h, as previously reported in the procedure. 4-ethyl benzoyl chloride (8.32 mmol) was added to an ice-cold solution of *N*-amino salt (4.16 mmol) in anhydrous tetrahydrofuran (40 mL) containing trimethylamine (2.0 mL). The mixture was allowed to proceed for 12 h at 70 °C to obtain *N*-ylide as a stable crystalline solid. Sodium borohydride (50.0 mmol) reduction of ylide in absolute ethanol (50 mL) furnished the 4-Ethyl-*N*-(7-hydroxy-3,4-dihydroisoquinolin-2(1H)-yl) benzamide (GM-4-53), with fair to good yield (65% yield).

### 5.2. Cell Culture

In this study, TNBC MDA-MB-231 cells (ATCC^®^ HTB-26™) were obtained from American Type Culture Collection (Manassas, VA, USA). MDA-MB-231 cells were initially brought up in ATCC-formulated Leibowitz’s L-15 medium, supplemented with 10% fetal bovine serum (FBS) and penicillin/streptomycin (100 U/0.1 mg/mL). After confluence, the cells were sub-cultured and grown in Dulbecco’s modified Eagle’s medium (DMEM), containing phenol red, 7.5% FBS, 4 mM L-glutamine, 20 µM sodium pyruvate, and penicillin/streptomycin (100 U/0.1 mg/mL). Cells were maintained at 37 °C in 5% CO_2_/atmosphere and, every 2–5 days, the medium was replaced, and cells were sub-cultured.

The anti-proliferative activity of GM-4-53 on two additional cell lines was evaluated at the Southern Research Institute (SRI, Birmingham, Alabama, USA). There, the human MCF-7 breast cancer cell line was purchased from the NCI. The human Ishikawa endometrial cancer cell line was purchased from Sigma Aldrich. Both cell lines were cultured in phenol red-free RPMI-1640 (Hyclone) (500 mL), supplemented with L-glutamine-dipeptide (Hyclone) (5 mL), and 10% fetal bovine serum (Atlanta Biologicals) (50 mL).

### 5.3. Proliferation and Cell Viability Studies

For MDA-MB-231 cells, both paclitaxel and GM-4-53 were dissolved in DMSO and stored at −20 °C. A stock solution for both compounds was prepped in HBSS. In brief, 96-well plates were seeded with cells (0.04 × 10^5^/well) to a volume of 200 µL, to which drugs were added, and cell proliferation (as cell count) was evaluated at a 6-day endpoint, using Alamar blue. For Alamar blue testing, a working solution of resazurin prepared in sterile phosphate-buffered saline (PBS)-phenol red (0.5 mg/mL) was added (15% *v*/*v*) to each sample in a 96-well plate. Samples were returned to the incubator for 4–6 h, and the reduction in the dye (to resorufin, a fluorescent compound) caused by viable cells was quantitatively analyzed using a Synergy HTX multi-mode reader (Bio-Tek, Winooski, VT, USA), with excitation/emission wavelength settings at 550 nm/580 nm. Cell count was evaluated as % control, and the inhibitory growth (IG_50_) value was calculated by regression analysis. Cytotoxicity studies were carried out at equal dose concentrations as proliferation studies at a higher plating density (0.5 × 10^5^/well), over 24–36 h. The cytostatic effects of MCF-7 and Ishikawa cell lines were determined using the CellTiter-Glo (CTG) luminescent cell viability assay, which is based on the direct determination of intracellular ATP level (determining the number of viable cells in culture based on the quantification of ATP present), which signals the presence of metabolically active cells. Luminescence results were read on TriLux Luminometer. Optimal concentrations for microarray were established by a lack of cell toxicity, at a concentration showing a complete blockage of cell proliferation; paclitaxel (1 μg/mL) and GM-4-53 (5 µg/mL). For microarray studies, cells were plated and treated in 75 cm3 flasks for 36 h.

### 5.4. 3D Tumor Studies

MDA-MB-231 cells were plated in low-adhesion, round-bottom, sterile, 96-well plates at a cell density of (0.2 × 10^5^/well), in 200 uL of cell culture media centrifuged at 1800× *g* for 3 min prior to incubation at 37 °C in 5% CO_2_/atmosphere. For the first 3 days, cells were taken out and re-centrifuged at 1800× *g* × 3 min. 3D spheroids were left to grow for 10 days. On day 14, experimental drugs were added, and images were captured on day 24.

### 5.5. Imaging

2D cell cultures were imaged to ascertain changes in morphological structure at the 6-day endpoint in MDA-MB-231 cells in the presence or absence of treatments. In brief, cells were fixed in 4% paraformaldehyde and incubated at 37 °C in 5% CO_2_/atmosphere for 15 min. After delicate removal of paraformaldehyde, sterile ultra-pure biological grade water containing 0.2% Triton X-100 was gently added to each well (100 uL) and returned to the incubator for 45 min. After gentle removal, a PBS solution containing propidium iodide (1 μg/mL) and Phalloidin-iFluor 488 Reagent (ab176753) was added to each well, according to the manufacturer’s instructions Abcam (Cambridge, MA, USA), and images were captured on an inverted fluorescent microscope using the 10× and 25× fluo-objectives. 3D tumors were stained with fluorescein diacetate (live-cell dye) and countered imaged for morphology.

### 5.6. Microarray WT 2.1 Human Datasets

After the experimental 36-h time point, the cells were scraped, washed three times in ice-cold HBSS, and spun down. The supernatant was removed, and the remaining pellet was rapidly frozen and stored at −80 °C. Total RNA was isolated and purified using the Trizol/chloroform method. The RNA quality was assessed, and concentrations were equalized to 82 ng/μL in nuclease-free water. According to the GeneChipTM WT PLUS Reagent Manual for Whole Transcript (WT) expression arrays, whole-transcriptome analysis was conducted. In brief, RNA was reverse-transcribed to first-strand/second-strand cDNA, followed by cRNA amplification and purification. After the 2nd cycle of ss-cDNA Synthesis and hydrolysis of RNA, ss-cDNA was assessed for yield, fragmented, labeled, and hybridized onto the arrays before being subjected to fluidics and imaging using the Gene Atlas Affymetrix, ThermoFisher Scientific (Waltham, MA, USA). The array data, quality control, and initial processing from CEL to CHP files were conducted using an expression console before data evaluation using the Affymetrix transcriptome analysis console (Wiki/Kegg pathways) and protein-protein interaction (PPI) String Database (String Consortium 2020) https://string-db.org, accessed on 25 March 2021 [98,99], *n* = 3.

### 5.7. Data Analysis

Statistical analysis was performed using Graph Pad Prism (San Diego, CA, USA). The significance of the difference between groups was assessed using a one-way ANOVA, followed by the Tukey post hoc means comparison test or Student’s *t*-test. IG_50_s were determined by regression analysis using origin software Origin Lab (Northampton, MA, USA).

## Figures and Tables

**Figure 1 ijms-22-07694-f001:**
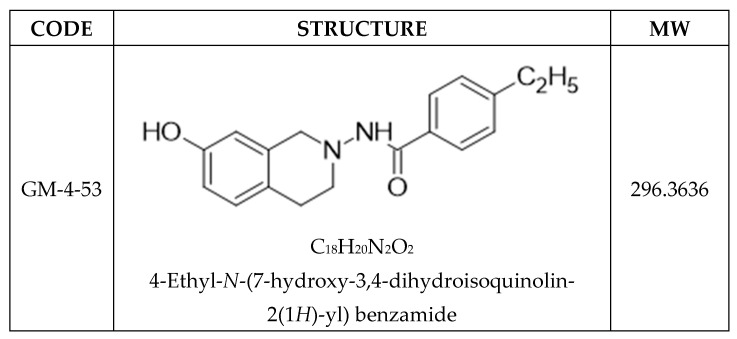
Chemical Structure of GM-4-53.

**Figure 2 ijms-22-07694-f002:**
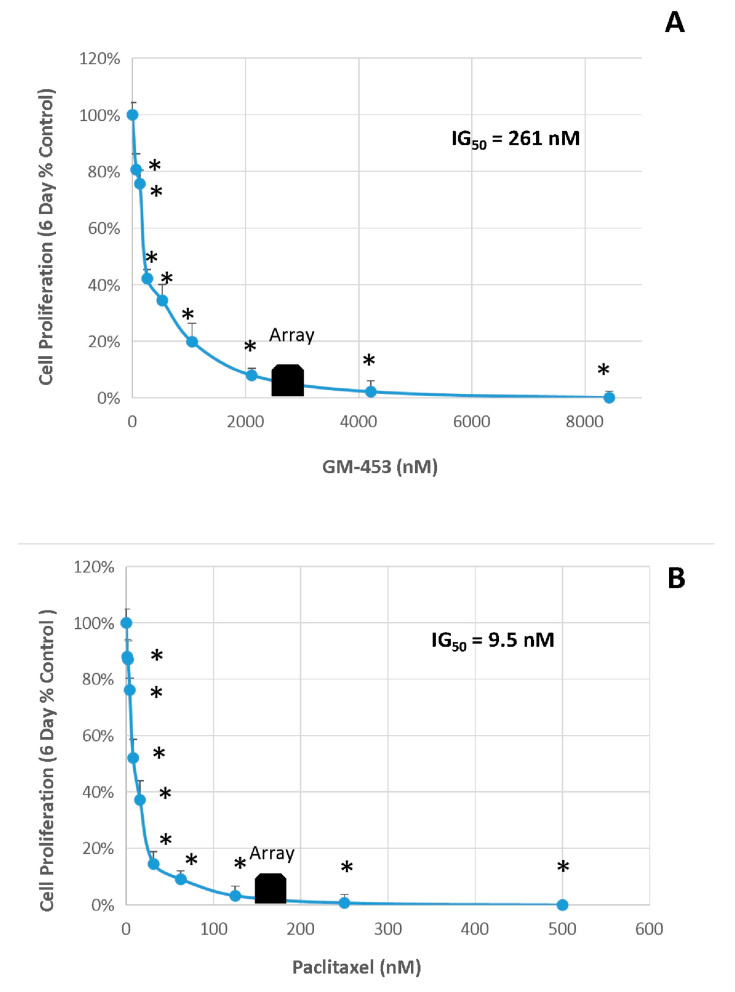
Cytostatic growth-inhibitory effects of GM-4-53 (**A**) and paclitaxel (**B**) in MDA-MB-231 cells were determined with an endpoint cell count analysis conducted on day 6. The data represent cell count/proliferation (as % untreated control) and are expressed as the Mean ± SEM, with the IG_50_s determined by regression analysis. Significant differences from the control were determined using a one-way ANOVA followed by a Tukey post hoc test. * *p* < 0.05. Array blocks denote the specific concentrations used for microarray analysis vs. controls.

**Figure 3 ijms-22-07694-f003:**
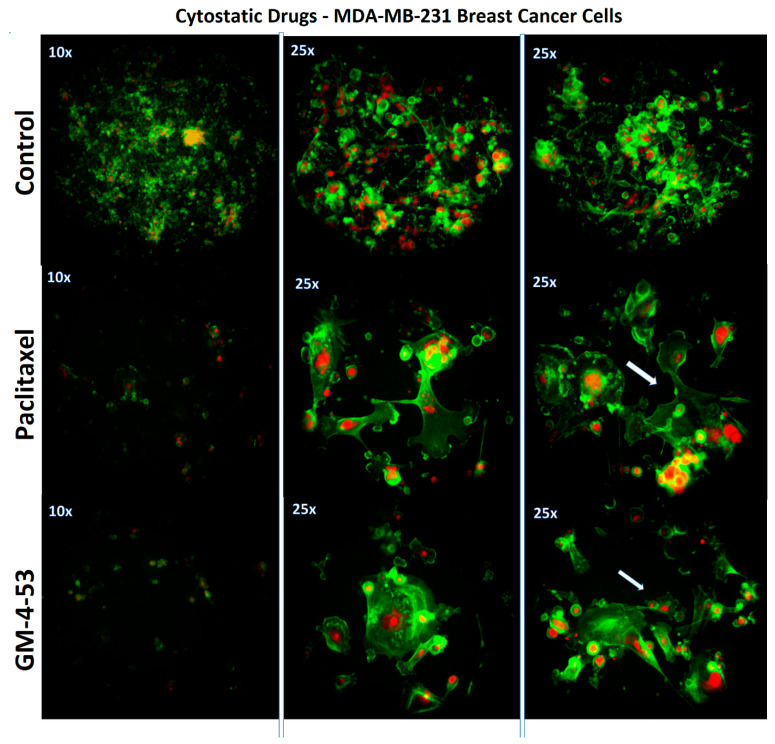
Morphological Changes associated with cytostatic growth-inhibitory effects by GM-4-53 5 μg/mL and paclitaxel 1 μg/mL in MDA-MB-231 cells evaluated at day 6. The images demonstrate cytostatic changes to the actin cytoskeletal architecture (green) with a propidium iodide nuclear counterstain (red) in the presence of either drug vs. untreated controls. [10×] shows the basic overview of accumulated cell number (cell proliferation), where [25×] shows higher magnification to enable visualization of deformed cell morphology *n* = 2. The images show abnormally large-shaped extended actin networks in cells that are unable to divide.

**Figure 4 ijms-22-07694-f004:**
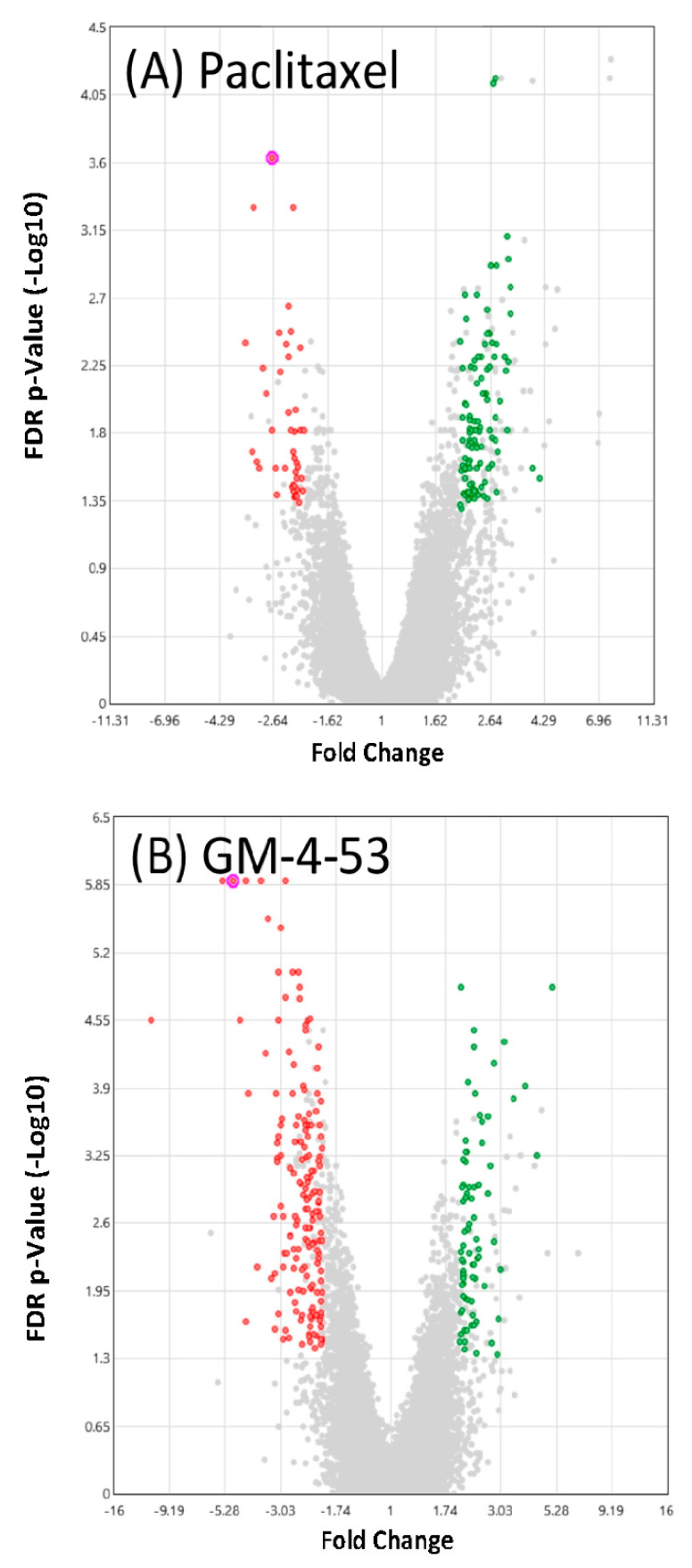
Summary of gene changes by fold change and significance in cytostatic cells vs. untreated controls from a total of 48,226 genes analyzed. (**A**) paclitaxel (1 μg/mL) vs. controls (**B**) GM 4-53 (5 μg/mL) in MDA-MB-231. The data show upregulated DEGs (right/green) and downregulated DEGs (left/red) by fold change relative to untreated controls (*X*-axis), with FDR *p*-values (*Y*-axis). Selected criteria for array analysis: Fold change >2 or <−2, *p*-value < 0.05, and false discovery rate (FDR) *p*-value < 0.05. Plotted points denoted in gray did not meet the selected criteria.

**Figure 5 ijms-22-07694-f005:**
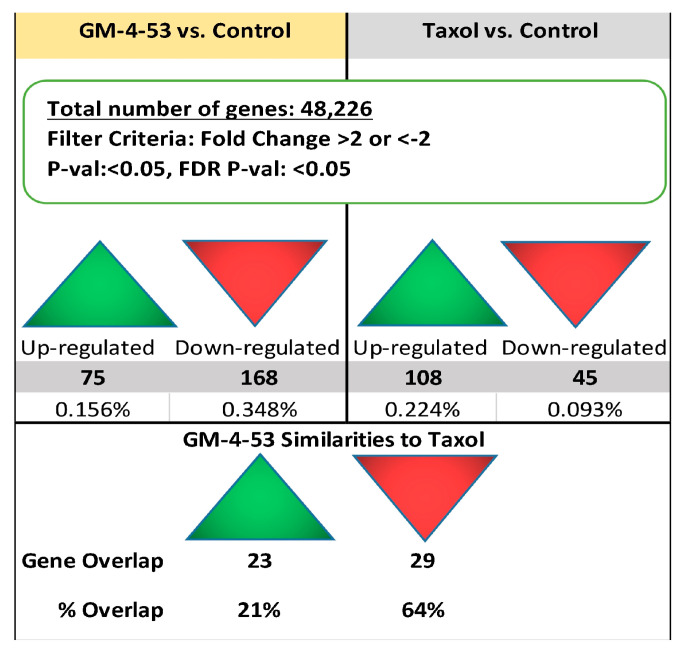
Gene Summary Report. Overall summary of DEGs reflected by whole-transcriptomic analysis in GM-4-53 (5 μg/mL)- or Paclitaxel (1 μg/mL)-treated cells vs. controls after 36 h of incubation in MDA-MB-231 cells; Criteria = FC > 2 or <−2, *p*-Value < 0.05 and FDR *p*-value < 0.05. Information on overlapping DEGs common to paclitaxel and GM-4-53 are also displayed. The gene summary data for each DEG, along with gene symbol, description, fold change, and significance, are presented in Appendix A.

**Figure 6 ijms-22-07694-f006:**
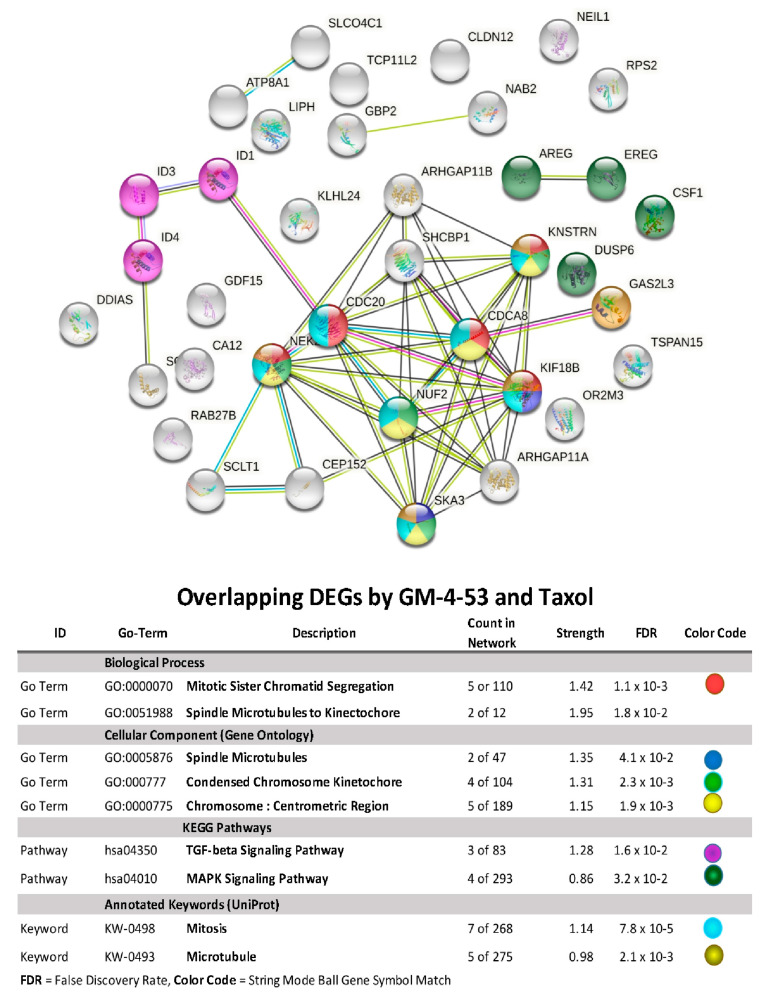
Functional analysis of the 52 transcripts overlaps in DEGs by paclitaxel and GM-4-53 vs. controls. The data represent significant functional changes as identified by several databases: local network cluster (STRING), Kegg pathways, Reactome Pathways expected protein domains and feature (Interpore), where data are reflected by count in network, strength, and false discovery rates.

**Figure 7 ijms-22-07694-f007:**
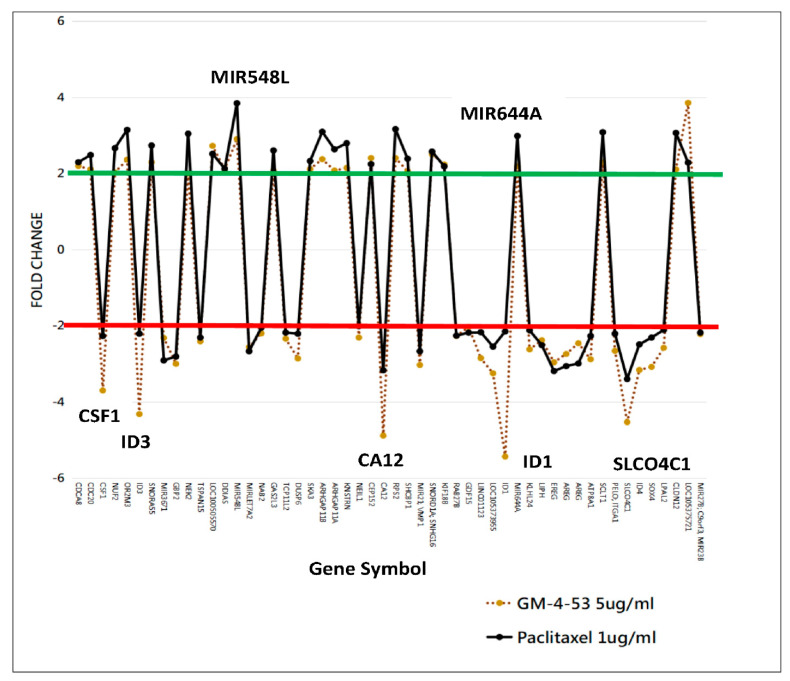
Overlapping DEG profile between GM-4-53 and paclitaxel. The data represent fold change for genes meeting the selection criteria = FC > 2 or <−2, *p*-value < 0.05, and FDR *p*-value < 0.05.

**Figure 8 ijms-22-07694-f008:**
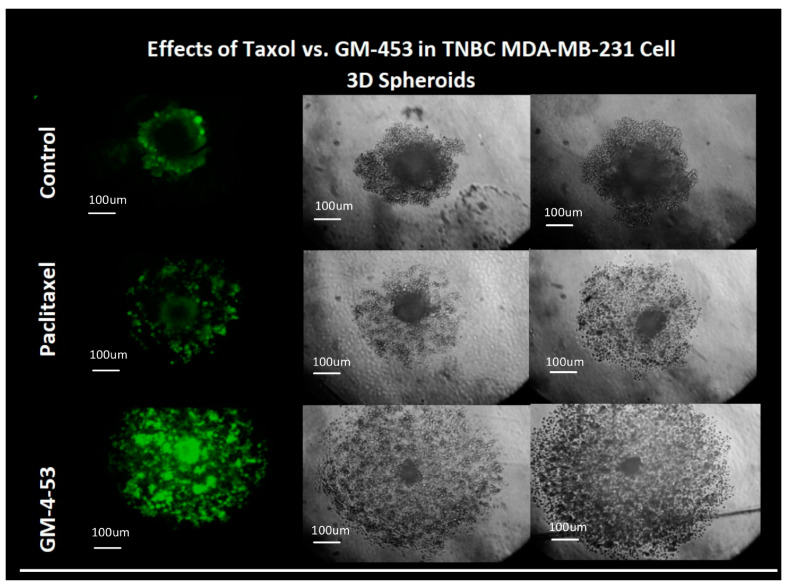
Effect of GM-4-53 (5 μg/mL) and paclitaxel (1 μg/mL) on 3D tumors. The left panel represents cell viability (fluorescein diacetate), and the right 2 panels show basic cell morphology (*n* = 2).

**Table 1 ijms-22-07694-t001:** Variable cell line: cytostatic growth-inhibitory effects of GM-4-53 in Ishikawa, MCF-7 vs. MDA-MB-231 cells were determined by regression analysis. The data show a consistent cytostatic effect in diverse female cancers in the mid-to-high nM range.

	MCF-7	Ishikawa	MDA-MB-231
	Human Breast Cancer (ER+, PR+)	Human Endometrial Cancer	Human Breast Cancer (TNBC) (ER_−_) (PR_−_)
	IG50 (nM)	IG50 (nM)	IG50 (nM)
GM-4-53	674.01	269.61	261.26

**Table 2 ijms-22-07694-t002:** Overlapping DEG profiles between GM-4-53 and paclitaxel. The data represent fold change for genes meeting the selection criteria = FC > 2 or < −2, *p*-value < 0.05, and FDR *p*-value < 0.05.

		Paclitaxel			GM-4-53		
Symbol	Description	FC	*p*-Value	FDR *p*-Value	FC	*p*-Value	FDR *p*-Value
AREG	amphiregulin	−3.06	1.0 × 10^−4^	2.5 × 10^−2^	−2.74	1.0 × 10^−4^	1.2 × 10^−2^
ARHGAP11A	Rho GTPase activating protein 11A	2.64	3.3 × 10^−6^	1.2 × 10^−3^	2.08	3.3 × 10^−6^	1.1 × 10^−3^
ARHGAP11B	Rho GTPase activating protein 11B	3.10	6.8 × 10^−6^	5.3 × 10^−3^	2.38	3.9 × 10^−5^	5.5 × 10^−3^
ATP8A1	ATPase, (APLT), class I, 8A, m1	−2.27	2.2 × 10^−6^	3.3 × 10^−3^	−2.88	4.9 × 10^−9^	1.7 × 10^−5^
CA12	carbonic anhydrase XII	−3.17	8.6 × 10^−8^	5.0 × 10^−4^	−4.88	3.0 × 10^−11^	1.3 × 10^−6^
CDC20	cell division cycle 20	2.49	3.0 × 10^−4^	4.2 × 10^−2^	2.11	7.0 × 10^−4^	3.5 × 10^−2^
CDCA8	cell division cycle associated 8	2.30	6.7 × 10^−5^	1.9 × 10^−2^	2.20	3.2 × 10^−5^	4.9 × 10^−3^
CEP152	centrosomal protein 152 kDa	2.25	2.0 × 10^−4^	3.4 × 10^−2^	2.41	3.6 × 10^−5^	5.2 × 10^−3^
CLDN12	claudin 12	3.07	1.6 × 10^−7^	8.0 × 10^−4^	2.11	5.0 × 10^−6^	1.4 × 10^−3^
CSF1	colony stimulating factor 1	−2.27	2.1 × 10^−6^	3.3 × 10^−3^	−3.70	8.8 × 10^−11^	1.3 × 10^−6^
DDIAS	DNA damage-induced apoptosis suppressor	2.13	1.6 × 10^−6^	2.8 × 10^−3^	2.11	7.1 × 10^−7^	4.0 × 10^−4^
DUSP6	dual specificity phosphatase 6	−2.21	9.2 × 10^−8^	5.0 × 10^−4^	−2.86	8.8 × 10^−11^	1.3 × 10^−6^
EREG	epiregulin	−3.19	8.6 × 10^−5^	2.1 × 10^−2^	−2.96	9.4 × 10^−6^	2.2 × 10^−3^
GAS2L3	growth arrest-specific 2 like 3	2.61	7.9 × 10^−6^	5.8 × 10^−3^	2.08	5.9 × 10^−5^	7.3 × 10^−3^
GBP2	guanylate binding protein 2, interferon-inducible	−2.81	1.7 × 10^−5^	8.6 × 10^−3^	−3.00	1.2 × 10^−6^	6.0 × 10^−4^
GDF15	growth differentiation factor 15	−2.18	4.0 × 10^−4^	4.2 × 10^−2^	−2.03	6.0 × 10^−5^	7.3 × 10^−3^
ID1	inhibitor of DNA binding 1, dom-HLHP	−2.15	2.0 × 10^−4^	3.2 × 10^−2^	−5.43	1.3 × 10^−10^	1.3 × 10^−6^
ID3	inhibitor of DNA binding 3, dom-HLHP	−2.21	8.5 × 10^−4^	2.1 × 10^−2^	−4.32	1.4 × 10^−10^	1.3 × 10^−6^
ID4	inhibitor of DNA binding 4, dom-HLHP	−2.49	1.0 × 10^−5^	6.2 × 10^−3^	−3.16	1.4 × 10^−7^	1.0 × 10^−4^
KIF18B	kinesin family member 18B	2.19	2.8 × 10^−5^	1.2 × 10^−2^	2.24	5.2 × 10^−6^	1.4 × 10^−3^
KLHL24	kelch-like family member 24	−2.12	1.0 × 10^−4^	2.7 × 10^−2^	−2.62	7.6 × 10^−7^	4.0 × 10^−4^
KNSTRN	kinetochore-loc. astrin/SPAG5bp	2.80	3.6 × 10^−7^	1.2 × 10^−3^	2.15	4.5 × 10^−6^	1.3 × 10^−3^
LIPH	lipase, member H	−2.51	2.5 × 10^−6^	3.4 × 10^−3^	−2.38	8.8 × 10^−7^	5.0 × 10^−4^
LPAL2	lipoprotein, Lp(a)-like 2, pg	−2.11	4.0 × 10^−4^	4.6 × 10^−2^	−2.58	1.3 × 10^−5^	2.6 × 10^−3^
NAB2	NGFI-A binding protein 2 (EGR1 binding protein 2)	−2.05	3.0 × 10^−4^	3.9 × 10^−2^	−2.20	8.6 × 10^−6^	2.0 × 10^−3^
NEIL1	nei-like DNA glycosylase 1	−2.01	4.7 × 10^−5^	1.5 × 10^−2^	−2.31	6.5 × 10^−7^	4.0 × 10^−4^
NEK2	NIMA-related kinase 2	3.05	4.7 × 10^−5^	1.5 × 10^−2^	2.00	7.0 × 10^−4^	3.5 × 10^−2^
NUF2	NUF2, NDC80 kinetochore com c	2.67	5.9 × 10^−5^	1.7 × 10^−2^	2.03	5.0 × 10^−4^	2.9 × 10^−2^
PELO; ITGA1	pelota hom (Dros); integrin alpha 1	−2.21	2.0 × 10^−4^	3.5 × 10^−2^	−2.66	2.3 × 10^−6^	8.0 × 10^−4^
RAB27B	RAB27B, member RAS oncogene family	−2.26	4.1 × 10^−5^	1.5 × 10^−2^	−2.27	1.1 × 10^−6^	6.0 × 10^−4^
RPS2	ribosomal protein S2	3.17	5.4 × 10^−7^	1.7 × 10^−3^	2.41	3.3 × 10^−6^	1.1 × 10^−3^
SCLT1	sodium channel/clathrin linker 1	3.09	2.7 × 10^−7^	1.1 × 10^−3^	2.30	3.5 × 10^−6^	1.1 × 10^−3^
SHCBP1	SHC SH2-domain binding protein 1	2.39	5.1 × 10^−5^	1.6 × 10^−2^	2.06	6.4 × 10^−5^	7.7 × 10^−3^
SKA3	Spindle & Kinetochore complex s3	2.33	1.0 × 10^−4^	2.7 × 10^−2^	2.12	2.0 × 10^−4^	1.3 × 10^−2^
SLCO4C1	SC organic anion transporter fm4C1	−3.40	3.7 × 10^−6^	4.0 × 10^−3^	−4.53	1.2 × 10^−8^	2.8 × 10^−5^
SOX4	SRY box 4	−2.31	5.4 × 10^−6^	4.9 × 10^−3^	−3.08	2.0 × 10^−9^	9.6 × 10^−6^
TSPAN15	tetraspanin 15	−2.31	1.1 × 10^−6^	2.3 × 10^−3^	−2.41	9.7 × 10^−8^	1.0 × 10^−4^

## Data Availability

The dataset has been deposited to NIH Gene Expression Omnibus located at https://www.ncbi.nlm.nih.gov/geo/query/acc.cgi?acc=GSE163046. Published on 12 December 2020.

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
