# Peer review of "Transcriptome Profile Analysis of Triple-Negative Breast Cancer Cells in Response to a Novel Cytostatic Tetrahydroisoquinoline Compared to Paclitaxel"

_ijms, 2021, doi:10.3390/ijms22147694_

Round 1

Reviewer 1 Report

The authors have significantly improved the manuscript and could be accepted for publication.

Author Response

Reviewer 1: 

The manuscript named "Transcriptome profile analysis of triple-negative breast cancer 2 cells in response to the novel anti-mitotic tetrahydroisoquinoline compound relative to paclitaxel" deals with a highly important topic – triple-negative breast cancer.

  • Comment: The manuscript is designed well. However, there are many shortcomings that should be fixed. F. e. line 55, after bracket with citations, the dot is missing.

Response: The dot has been added after the citation bracket [], and the manuscript has been carefully edited. 

  • Comment: Figure 1 is shifted to the left.

Response: We have changed the indented format for Fig.1 to center its alignment within the text.

----------------------

  • Comment: In the text, once Fig. is mentioned (f. e. line 69), figure (f. e. line 146).

Response: "Figure" terminology has all been edited to "Fig." rather than figure throughout the manuscript.  

    ----------------------

  • Comment: Lines 155-156, there are many types of letters used, once bold, once normal. Please, unify it.

Response: The text in Lines 155-156 is now uniform.

          ----------------------

  • Comment: Line 159, the abbreviation DEGs was already explained, there is no need to explain it once more. On the other hand, in the figure legend, all abbreviations have to be explained, f. e. DEGs, because figure could stay alone without the text.

Response: We have changed all references to "DEGs" to appear in each legend first with description, then being referred to DEG for all subsequent instances.

----------------------

  • Comment: Figure legends should stay on the same page as figures, there is no need to have the figures as big as one A4.

Response: We have edited the text to where all figure legends are now on the same page as the figures.

----------------------

  • Comment: The discussion part is written well. However, there are many abbreviations that are not explained at all, and stay in the discussion for the first time, f.e. OSM, ID1, etc. Once, the authors write the names of drugs with a big letter at the beginning, once normal. Please, unify!

Response: Descriptive associations for abbreviations have been corrected for each first instance of appearing in the text. Drugs names are now uniform.

----------------------

  • Comment: Line 259 – wrong citations

Response: This citation has been corrected.   

----------------------

Reviewer 1: 

The manuscript named "Transcriptome profile analysis of triple-negative breast cancer 2 cells in response to the novel anti-mitotic tetrahydroisoquinoline compound relative to paclitaxel" deals with a highly important topic – triple-negative breast cancer.

  • Comment: The manuscript is designed well. However, there are many shortcomings that should be fixed. F. e. line 55, after bracket with citations, the dot is missing.

Response: The dot has been added after the citation bracket [], and the manuscript has been carefully edited. 

  • Comment: Figure 1 is shifted to the left.

Response: We have changed the indented format for Fig.1 to center its alignment within the text.

----------------------

  • Comment: In the text, once Fig. is mentioned (f. e. line 69), figure (f. e. line 146).

Response: "Figure" terminology has all been edited to "Fig." rather than figure throughout the manuscript.  

    ----------------------

  • Comment: Lines 155-156, there are many types of letters used, once bold, once normal. Please, unify it.

Response: The text in Lines 155-156 is now uniform.

          ----------------------

  • Comment: Line 159, the abbreviation DEGs was already explained, there is no need to explain it once more. On the other hand, in the figure legend, all abbreviations have to be explained, f. e. DEGs, because figure could stay alone without the text.

Response: We have changed all references to "DEGs" to appear in each legend first with description, then being referred to DEG for all subsequent instances.

----------------------

  • Comment: Figure legends should stay on the same page as figures, there is no need to have the figures as big as one A4.

Response: We have edited the text to where all figure legends are now on the same page as the figures.

----------------------

  • Comment: The discussion part is written well. However, there are many abbreviations that are not explained at all, and stay in the discussion for the first time, f.e. OSM, ID1, etc. Once, the authors write the names of drugs with a big letter at the beginning, once normal. Please, unify!

Response: Descriptive associations for abbreviations have been corrected for each first instance of appearing in the text. Drugs names are now uniform.

----------------------

  • Comment: Line 259 – wrong citations

Response: This citation has been corrected.   

----------------------

Reviewer 2 Report

The article from Gangapuram and colleagues entitled “Transcriptome profile analysis of triple-negative breast cancer cells in response to the novel antimitotic tetrahydroisoquinoline compound relative to paclitaxel” presents the similarities in gene expression regulation between MM231 breast cancer cells treated with Paclitaxel and GM-453, the overlap between affected genes across these treatments, and the pathway these relate to.

Overall the study is well executed and the data is presented clearly. The main issue here is reproducibility. First, I cannot seem to find the information about how many replicate analyses were performed. Second, analyzing only MM231 cells is reductive and does not take into account model-dependent changes. A second cell line should be included here so to give the conclusions a bit more credibility. On top of this, validation of gene downregulation for (e.g.) TOP10 DEGs should be performed by PCR analysis.

  • Figure 3 has low resolution, please provide an image with higher DPI or a vector-graphic based one.
  • Figure 4: x-axes on volcanoes seem squeezed. Please amend.

Author Response

Reviewer 2: Comments and Suggestions for Authors

Transcriptome profile analysis of triple-negative breast cancer cells in response to the novel anti-mitotic tetrahydroisoquinoline compound relative to paclitaxel

  • Comment: In this paper, the authors investigated the potential of the novel anti-mitotic compound in triple-negative breast cancer cells. They tried proliferation studies and some transcriptome analysis to estimate the potential compared with paclitaxel. It is interesting that they found some useful drug for TNBC chemotherapy, but I think the evidence of this study are too weak to conclude as authors mentioned. I also have several minor points in the figure which I could not understand. Why authors chose paclitaxel for comparison with GM-4-53? More information regarding to this point should be added in the introduction part.

Response:  We have elaborated more on paclitaxel, its mechanism of action, and its relevance to the work throughout the manuscript, primarily in the discussion.    

  • Comment: Authors only checked the evidence from transcriptome analysis in a cell line with proliferation assay. It could explain GM-4-53 has an anti-mitotic effect in breast cancer, but other signatures like anti-stromal/fibrotic, anti-TAM recruitment, etc. have not been checked in vitro.

Response:   We have added Table 1 with additional data - comparing three female cancer cell lines, where GM-4-53 is reasonably consistent in establishing cytostatic effects in the mid to high nM range.  The compound was sent out for 3rd party testing to Southern Research Institute. We tested this drug for anti-inflammatory effects in activated macrophages but did not find any effect.

  • Comment: Across the study, the authors emphasized the commonality of two drugs, GM-4-53, and paclitaxel. 72 common DEGs of about 200 DEGs in each group, however, is less than a half and not quite a large proportion, I think.

Response: We reanalyzed the study to tighten up significance to p and FDR values being less than .05. With this, we found only 52 overlaps, most of these involving cell cycle, spindle, and chromosomal separation. 

  • Comment: What is the "arrow" box means in Figure 1? I think it is a kind of mistake in visualization.

Response: This was an error and has been corrected.

  • Comment: In Figure 4, Why are some dots colored in gray, even though they have such fold change value and FDR p-value?

Response: The legend has been corrected to identify gray data points as not meeting the criteria of -2<>2, p-Value <.05.

Round 2

Reviewer 2 Report

n/a

This manuscript is a resubmission of an earlier submission. The following is a list of the peer review reports and author responses from that submission.

Round 1

Reviewer 1 Report

The manuscript named „Transcriptome profile analysis of triple-negative breast cancer 2 cells in response to the novel anti-mitotic tetrahydroisoquinoline compound relative to paclitaxel” deals with a highly important topic – triple-negative breast cancer.

The manuscript is designed well. However, there are many shortcomings that should be fixed. F. e. line 55, after bracket with citations, the dot is missing. Figure 1 is shifted to the left. In the text, once Fig. is mentioned (f. e. line 69), once Figure (f. e. line 146). Please, use one uniform form throughout the manuscript. Lines 155-156, there are many types of letters used, once bold, once normal. Please, unify it. Line 159, the abbreviation DEGs was already explained, there is no need to explain it once more. On the other hand, in the figure legend, all abbreviations have to be explained, f. e. DEGs, because figure could stay alone without the text. Figure legends should stay on the same page as figures, there is no need to have the figures as big as one A4.

The discussion part is written well, however, there are many abbreviations that are not explained at all, and stay in the discussion for the first time, f.e. OSM, ID1, etc. Once, the authors write the names of drugs with a big letter at the beginning, once normal. Please, unify!

Line 259 – wrong citations.

Author Response

Reviewer 1: Comments and Suggestions for Authors

The manuscript named „Transcriptome profile analysis of triple-negative breast cancer 2 cells in response to the novel anti-mitotic tetrahydroisoquinoline compound relative to paclitaxel” deals with a highly important topic – triple-negative breast cancer.

Comment: The manuscript is designed well. However, there are many shortcomings that should be fixed. F. e. line 55, after bracket with citations, the dot is missing.

Response: The manuscript has been carefully edited, and all revisions are in the document (with track changes settings turned on). The suggested corrections have been made.

Comment: Figure 1 is shifted to the left.

Response: We have changed the indent format for Figure 1 to center it in alignment within the text.

Comment: In the text, once Fig. is mentioned (f. e. line 69), once Figure (f. e. line 146).

Response: All Figures are now denoted consistently through out the text as “Fig.”

Comment: Lines 155-156, there are many types of letters used, once bold, once normal. Please, unify it.

Response: This correction has been made

Comment: Line 159, the abbreviation DEGs was already explained, there is no need to explain it once more. On the other hand, in the figure legend, all abbreviations have to be explained, f. e. DEGs, because figure could stay alone without the text.

Response: We have changed all references to “DEGs” to appear in each legend first with description , then referred to DEG for all subsequent instances .   

Comment: Figure legends should stay on the same page as figures, there is no need to have the figures as big as one A4.

Response: We are requesting for assistance by the editors in the typesetting of this particular issue in the document.

Comment: The discussion part is written well, however, there are many abbreviations that are not explained at all, and stay in the discussion for the first time, f.e. OSM, ID1, etc. Once, the authors write the names of drugs with a big letter at the beginning, once normal. Please, unify!

Response: Descriptive association for abbreviations have been corrected for each first instance of appearing in the text. Drugs names are now uniform.  

Comment : Line 259 – wrong citations.

Response: This error has been located and fixed.

Reviewer 2 Report

Transcriptome profile analysis of triple-negative breast cancer cells in response to the novel anti-mitotic tetrahydroisoquinoline compound relative to paclitaxel

In this paper, authors investigated the potential of the novel anti-mitotic compound in triple-negative breast cancer cells. They tried proliferation studies and some transcriptome analysis to estimate the potential compared with paclitaxel. It is interesting that they found some useful drug for TNBC chemotherapy, but I think the evidences of this study are too weak to conclude as authors mentioned. I also have several minor points in figure which I couldn’t understand.

  1. Why authors chose paclitaxel for comparing with GM-4-53? More information regarding to this point should be added in introduction part.
  2. Authors only checked the evidences from transcriptome analysis in cell line with proliferation assay. It could explain GM-4-53 have anti-mitotic effect in breast cancer, but other signatures like anti-stromal/fibrotic, anti-TAM recruitment etc. haven’t been checked in vitro.
  3. Across the study, authors emphasized the commonality of two drugs, GM-4-53 and paclitaxel. 72 common DEGs of about 200 DEGs in each group, however, is less than a half and not quite large proportion, I think.
  4. What is “arrow” box means in Figure 1? I think it is kind of mistake in visualization.
  5. In Figure 4, Why some dots are colored in gray, even though they have such fold change value and FDR p-value?

Author Response

Reviewer 2: Comments and Suggestions for Authors

Transcriptome profile analysis of triple-negative breast cancer cells in response to the novel anti-mitotic tetrahydroisoquinoline compound relative to paclitaxel

Comment: In this paper, authors investigated the potential of the novel anti-mitotic compound in triple-negative breast cancer cells. They tried proliferation studies and some transcriptome analysis to estimate the potential compared with paclitaxel. It is interesting that they found some useful drug for TNBC chemotherapy, but I think the evidences of this study are too weak to conclude as authors mentioned. I also have several minor points in figure which I couldn’t understand.

Why authors chose paclitaxel for comparing with GM-4-53? More information regarding to this point should be added in introduction part.

Response: At the end of the introduction , we have added a bit more information on why we compare this drug to paclitaxel. The text reads as follows; “ Paclitaxel is a first in class antimitotic drug, and as one of the first taxanes to enter into clinical practice, remains clinically relevant today in combination with anthracy-clines and subject to advanced delivery systems such as nab-paclitaxel [38]. Given that TNBC is one of the most aggressive types of breast cancer, new drug therapies must continually be explored [39]. While, paclitaxel interferes with microtubule polymeriza-tion to induce cell cycle arrest [40-42], a novel synthetic compound; GM-4-53 (Fig. 1) while exerting similar effects in causing multi-nucleated severely arrested static cells, has a mechanism that does not involve polymerization or tubulin assembly [43]. In this study, we further investigate differences between GM-4-53 and paclitaxel in altering the whole transcriptome of TNBCs. “

Comment: Authors only checked the evidences from transcriptome analysis in cell line with proliferation assay. It could explain GM-4-53 have anti-mitotic effect in breast cancer, but other signatures like anti-stromal/fibrotic, anti-TAM recruitment etc. haven’t been checked in vitro.

Response: This is very true. The transcriptomic data only elude to DEGs being controlled by the drug that regulates these processes. We will be carrying out these studies in the near future to confirm possible therapeutic value of this drug. We have added a sentence to clarify this in the conclusion.

Comment: Across the study, authors emphasized the commonality of two drugs, GM-4-53 and paclitaxel. 72 common DEGs of about 200 DEGs in each group, however, is less than a half and not quite large proportion, I think.

Response: Yes this could be correct, but we emphasized this because rarely do we see two drugs with this level of overlap in our routine microarray data studies. We will modify our interpretation to be on the side of caution.  

Comment: What is “arrow” box means in Figure 1? I think it is kind of mistake in visualization.

Response: This has been corrected.

 Comment: In Figure 4, Why some dots are colored in gray, even though they have such fold change value and FDR p-value?

Response: The legend has been corrected to identify gray data points as not meeting the criteria of -2<>2, p-Value  <.05.  

Reviewer 3 Report

The article entitled 'Transcriptome profile analysis of triple-negative breast cancer cells in response to the novel anti-mitotic tetrahydroisoquino-line compound relative to paclitaxel' by Gangapuram et al. is an interesting work. The authors does explain nicely the potential of a new drug GM-4-53 for TNBC therapy. The quality of production was nice and it is scientifically sound. In my opinion the manuscript should be accepted for publication with minor changes.

Minor concerns:

Results:

Line 160 and 162: Do not abbreviate DEGs repeatedly. Do it the first time it appears and continue using the abbreviation.

 The authors could validate some genes by qRT-PCR. 

Author Response

Reviewer 3: Comments and Suggestions for Authors

The article entitled 'Transcriptome profile analysis of triple-negative breast cancer cells in response to the novel anti-mitotic tetrahydroisoquino-line compound relative to paclitaxel' by Gangapuram et al. is an interesting work. The authors does explain nicely the potential of a new drug GM-4-53 for TNBC therapy. The quality of production was nice and it is scientifically sound. In my opinion the manuscript should be accepted for publication with minor changes.

Minor concerns: Results:

Comment: Line 160 and 162: Do not abbreviate DEGs repeatedly. Do it the first time it appears and continue using the abbreviation.

Response: This has been corrected throughout the manuscript , also suggested by Reviewer #1. We have changed all DEGs in each legend to appear first with description , acronym and then DEG from then on.   

Comment: The authors could validate some genes by qRT-PCR.

Response: We will be conducting corroborating studies in the near future of major elements found in this work.